# The role of microstructure in the thermal fatigue of solder joints

J. W. Xian ®[1,2] ✉, Y. L. Xu ®[1,3] ✉, S. Stoyanov ®[4], R. J. Coyle[5], F. P. E. Dunne ®[1] & C. M. Gourlay ®[1] ✉

Thermal fatigue is a common failure mode in electronic solder joints, yet the role of microstructure is incompletely understood. Here, we quantify the evolution of microstructure and damage in Sn-3Ag-0.5Cu joints throughout a ball grid array (BGA) package using EBSD mapping of localised subgrains, recrystallisation and heavily coarsened $Ag_3Sn$. We then interpret the results with a multi-scale modelling approach that links from a continuum model at the package/board scale through to a crystal plasticity finite element model at the microstructure scale. We measure and explain the dependence of damage evolution on (i) the β-Sn crystal orientation(s) in single and multigrain joints, and (ii) the coefficient of thermal expansion (CTE) mismatch between tin grains in cyclic twinned multigrain joints. We further explore the relative importance of the solder microstructure versus the joint location in the array. The results provide a basis for designing optimum solder joint microstructures for thermal fatigue resistance.

There is an ongoing need to improve the understanding and reliability of Pb-free solders, particularly for applications with high consequences of failure in the medical, automotive, aerospace and defence industries. Thermal fatigue resistance is an important design requirement that can be approached through electronic package design optimisation and by the design of improved materials, including new solder alloys and the optimisation of solder microstructures.

A range of past studies on accelerated thermal cycling of Sn-Ag-Cu solder joints[1–4] have shown that damage accumulation is sensitive to the solder microstructure including tin crystal orientation(s) and the intermetallic length scale, to the extent that failure can occur in a joint that is far from the location of maximum global shear strain[1]. A key factor is that tetragonal β-Sn is highly anisotropic, for example having a coefficient of thermal expansion (CTE) and Young's modulus that are approximately two and three times larger respectively along the c crystallographic direction than along the a = b directions[5,6].

In the simplest case of single grain Sn-Ag-Cu solder joints, both experiments[1–3] and modelling[3,7–10] have shown that the most severe thermal fatigue damage develops in single grain joints when the β-Sn c-axis lies in the plane of the printed circuit board (PCB). This has been attributed to two factors[1,2]: (1) this orientation maximises the in-plane CTE mismatch between the solder and package and, therefore, maximises in-plane shear during temperature cycling and (2) this orientation also minimises the β-Sn CTE component normal to the PCB which induces a tensile component of stress during heating.

The case of multigrain BGA joints is more complex and the optimum microstructure for thermal fatigue resistance is unclear from past work. Bieler et al.[1] found that some multigrain joints in a BGA package were not cracked when subjected to thermal cycling while, at the same time, unfavourably oriented single grain joints were cracked, and discussed this in terms of the majority orientation of the β-Sn grains. Arfaei et al.[11] reported longer thermal fatigue life for joints containing β-Sn grains with interlaced morphology compared with beachball morphology. Various authors[12–15] have noted the importance of CTE mismatch at grain boundaries in damage development in multigrain joints, but these have not quantitatively linked the different orientations within joints to the local CTE mismatch and subsequent damage. At the same time, microstructurally motivated models have

[1]Department of Materials, Imperial College London, London, UK. [2]School of Materials Science and Engineering, Dalian University of Technology, Dalian, China. [3]Institute of High Performance Computing (IHPC), Agency for Science, Technology and Research (A*STAR), 1 Fusionopolis Way, #16-16 Connexis, Singapore, Republic of Singapore. [4]School of Computing and Mathematical Sciences, University of Greenwich, London, UK. [5]Nokia Bell Labs, Murray Hill, New Jersey, USA. ✉e-mail: jwxian@dlut.edu.cn; yilun.xu@imperial.ac.uk; c.gourlay@imperial.ac.uk

been developed[16–19] to investigate microstructural effects on failure mechanisms and damage evolution in joints subject to thermal cycling and/or shear stress. However, these have not directly compared experimental measurements of microstructure and damage in BGA joints with mechanistic three-dimensional (3D) models of the joints.

Therefore, there is a need for coupled experimental and modelling studies on thermal cycling of BGA packages including complex multigrain solder microstructures to build mechanistic understanding on the role of microstructure in thermal fatigue damage. This will provide a basis for designing optimum solder microstructures for long-term thermal reliability in microelectronics that can be implemented in parallel with solder alloy design approaches (e.g[20].).

In this work, we build mechanistic understanding of how the Sn-3Ag-0.5Cu (wt%) solder microstructure affects thermal fatigue resistance in a ball grid array (BGA) package. This paper contains three areas of advance in this direction that lead to insights into the role of microstructure in the thermal fatigue of BGA solder joints. (1) We develop an approach to processing electron backscatter diffraction (EBSD) maps that allows damage to be quantitatively compared across multiple complex Sn-Ag-Cu microstructure types, providing a versatile damage measure using the area fraction of the cross-section with misorientation larger than 8°. This approach is then applied to measure the damage accumulated in 20 time-zero joints and 84 thermally cycled joints from an 84CTBGA (thin chip array BGA) electronic test vehicle; (2) We develop a method to estimate the initial 3D microstructure of a BGA joint from a 2D EBSD map after mild thermal cycling; and (3) We perform a direct comparison between the experimentally measured microstructure and damage in selected BGA joints and a 3D multi-scale thermal cycling model of the same joints, coupling a continuum finite element (FE) model at the scale of the whole 84CTBGA package and board, with a crystal plasticity finite element (CPFE) model of individual solder joints. The approach is overviewed in Fig. 1. Detailed methods are given at the end of the manuscript.

## RESULTS

### Microstructure variability

In both time-zero and thermally cycled joints, the β-Sn microstructures were single grain or multigrain with partial interlacing. Some multigrain joints contained three main orientations related by cyclic twinning as has been reported widely for Sn-3Ag-0.5Cu joints[21–26]. Other multigrain joints contained five main orientations related by double rings of cyclic twinning, similar to references[27–30]. The β-Sn morphology types are compared in time-zero versus thermally cycled joints in

Fig. 2. Figure 2a, c are β-Sn orientation maps (inverse pole figure (IPF) maps with respect to the Y-direction) for time-zero joints and different thermally cycled joints respectively. The main β-Sn orientations are plotted as unit cell wireframes from the measured Euler angles and have been translated to highlight the cyclic twin relationships and the shared orientation in double rings. (b) and (d) are β-Sn misorientation (MO) maps, where the colour scale has been selected to clearly show small misorientations in the range 0−5° as changing shades of blue. Comparing the MO maps in Fig. 2b, d, note that the blue/white features are similar in the time-zero joints and the thermally cycled joints, both in terms of the microstructure revealed by the blue/white features and by the magnitude of the MO (the shade of blue). In contrast, there are regions with localised high MO near the top (the package side), and sometimes the bottom, in joints after thermal cycling (Fig. 2d) which are absent in all time-zero joints (Fig. 2b). These are regions of recrystallised grains, subgrains and local lattice curvature that are considered in more detail in the next section. From Fig. 2b, d, we see that strain and damage localise at the top and bottom of joints with less in central regions. Therefore, it is reasonable to use the major β-Sn orientation(s) from near the centre of a thermally cycled joint to infer the microstructure of that joint at time zero.

Supplementary Table 1 summarises the frequency of occurrence of β-Sn microstructure types for 20 time-zero joints and 84 thermally cycled joints, excluding recrystallised regions at the top and bottom. There is reasonable consistency in the percentage of each microstructure type in the time-zero and thermally cycled packages, although a relatively low number of time-zero joints were analysed (20). Supplementary Table 1 also shows that >96% of multigrain joints contained grains all interrelated through a twin relationship and, therefore, originated from a single β-Sn nucleation event[21].

From Supplementary Table 1 and Figs. 1–3, we see highly variable β-Sn microstructures from joint to joint. The origin of this variability is: (i) the stochastic undercooling for β-Sn nucleation[21–23,30,31], (ii) the variable β-Sn nucleation location, and (iii) the small number of grains developed in the small solder volume[1,16,24]. The effect of different Sn nucleation locations can be seen clearly by comparing the joints in Figs. 1d and 3a, where it can be inferred that nucleation occurred from the package (top) side in Fig. 3a, d and from the PCB (bottom) side in Fig. 1d.

In many cases, the 3D grain structure could be reasonably estimated from 2D EBSD maps. Our approach utilises the crystallographic geometry of the twin grain boundaries and is overviewed in Fig. 3. In Figs. 1–3, the multigrain joints contain some straight grain boundary

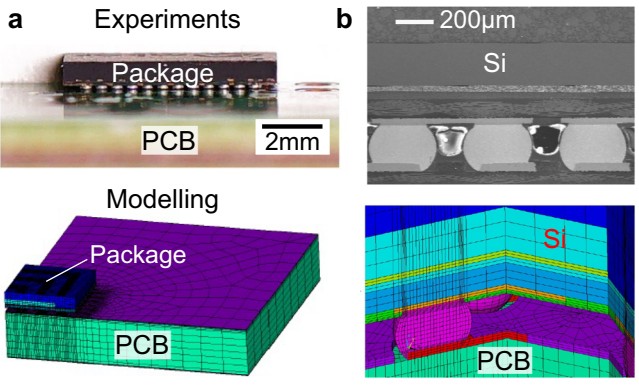
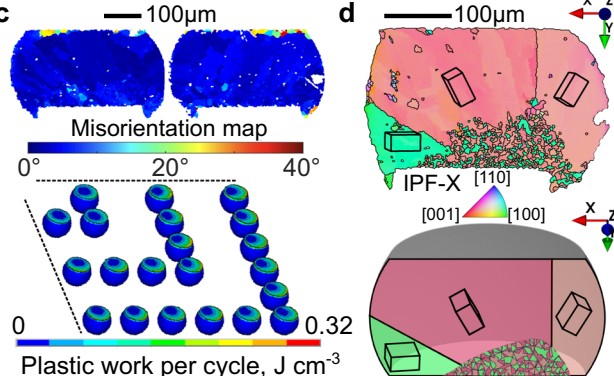

**Fig. 1 | Parallel thermal cycling experiments and modelling. a** 84CTBGA package soldered to a printed circuit board (PCB): photograph and quarter finite element model. **b** Micrograph and board-level finite element (FE) model of a cross-section through solder joints. **c** Damage accumulation in the solder near the package side after thermal cycling: experimental EBSD misorientation maps and accumulated plastic work per cycle from the board-level FE model. Displacement fields from the top and bottom of solder joints are then transferred from the board-level continuum model to the individual solder joint CPFE model. **d** Typical partially interlaced time-zero beachball microstructure from EBSD orientation mapping (Inverse Pole Figure IPF-X colour map) and 3D geometrical model input into the CPFE model.

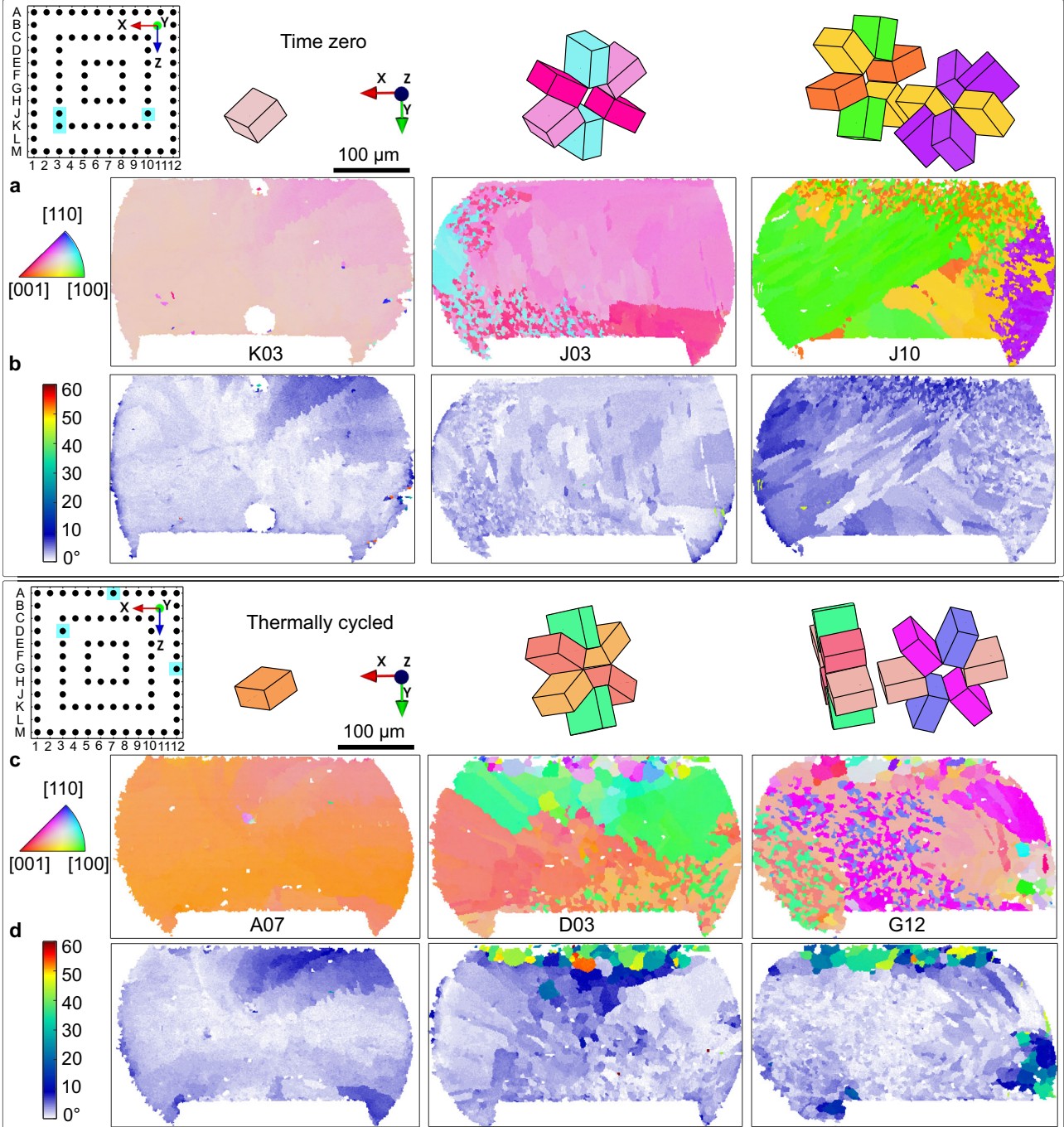

**Fig. 2 | Typical solder joint microstructures in the 84CTBGA before and after thermal cycling. a**, **b** Time-zero, **c**, **d** after 7580 thermal cycles from 0/100 °C. **a**, **c** EBSD orientation maps (IPF-Y) and Sn unit cell wireframes translated to highlight the cyclic twins. **b**, **d** EBSD misorientation maps with reference to the major Sn orientation(s). Pin diagrams indicate the locations of each joint in the array and labels of selected joints are marked under EBSD maps.

segments that are clearest away from regions of interlacing. Prior work by Lehman et al.[21] reported that such grain boundaries are {101} cyclic twin planes and result in "beachball" microstructures when the fraction of interlacing is low. Figure 3a, b shows this using one joint from this study and Fig. 3c confirms {101} macroscopic beachball boundaries from 29 joints reanalysed from EBSD datasets in[29,32]. Note in Fig. 3a that the boundaries are not straight {101} planes at the micrometre scale but can be well-approximated as {101} boundaries at the scale of the joint. Combining the measured cyclic twinned orientations, the measured 2D vectors formed where the grain boundaries intersect the polishing plane, and the knowledge that the grain boundaries are {101}

planes, enables the time-zero macroscopic 3D grain structure to be generated from measured 2D EBSD maps. This is demonstrated in Figs. 3d and 1d gives another example containing partial interlacing. Further details of the approach are given in the "Building 3D micro-structure models from 2D EBSD maps" part of the Methods section.

Compared with the tin grain structure, there was much less variability in eutectic intermetallic size among joints before thermal cycling and among those after thermal cycling. This is overviewed in Supplementary Fig. 1. There was also strong localised accelerated Ag$_3$Sn coarsening near the top interface, as is common in thermally cycled Sn-3Ag-0.5Cu joints[12,25,33] due to coarsening with both a

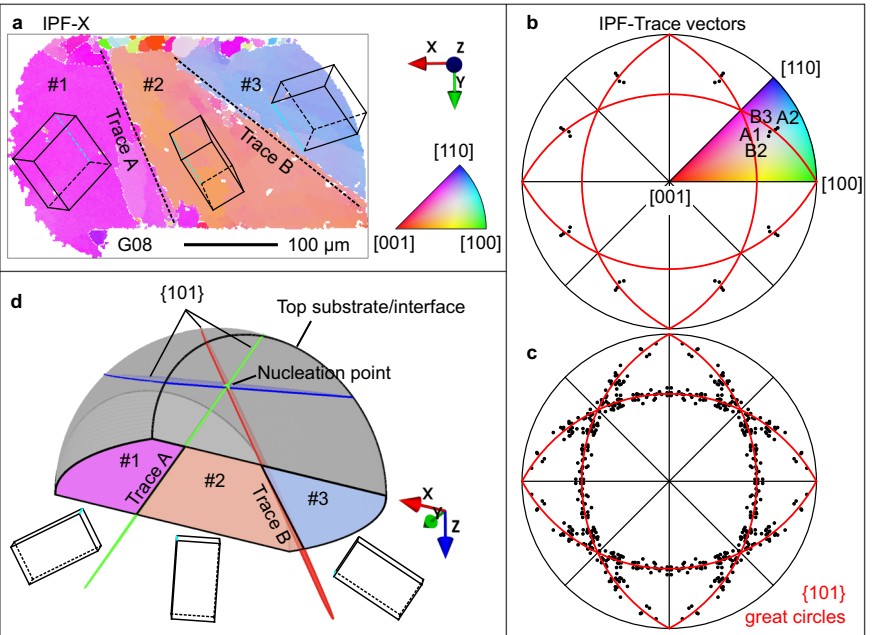

**Fig. 3 | 3D microstructure models from 2D EBSD maps. a** EBSD β-Sn orientation map with IPF-X colouring of a typical beachball microstructure with three twinned β-Sn grains whose 3D boundaries intersect the EBSD cross section forming near-linear 2D trace vectors A and B. **b** A stereogram plotting the 2D trace vectors A and B from (**a**) with respect to the β-Sn orientations on either side of Sn-Sn boundaries. The four datapoints (A1, A2, B2 and B3) are repeated 8 times according to the 4/mmm point group symmetry of β-Sn. **c** Summary of 82 IPF-trace vectors measured from 29 solder joints showing that macroscopic beachball grain boundaries are approximately {101}Sn planes. **d** A 3D digital model of the microstructure assuming that (**a**) was polished to the centre of the joint and that {101}Sn beachball boundaries emanate from a point where the common ⟨010⟩ twin axis touches the intermetallic reaction layer (the top side here). The 3D viewing direction in (**d**) is close to the ⟨010⟩ twin axis.

thermal[34–36] and strain-enhanced[12,37,38] component and further due to faster diffusion along recrystallised grain boundaries once recrystallisation has begun[12]. Here, these accelerated coarsened Ag₃Sn particles exceeded 2μm (Supplementary Fig. 1f) and are considered further in the next section.

## Microstructure evolution and damage metrics

The features of microstructure evolution in thermally cycled joints are overviewed in Fig. 4 using one joint as an example. Equiaxed recrystallised grains (with MO > 15°) have developed from the top-right corner and span ~80% of the way along the top region (Fig. 4b–d). To the left of and below the recrystallised grains are subgrains with MO < 15° (Fig. 4c, d) which appear as different shades of light blue in Fig. 4c. There is a strong correlation between recrystallised grains and accelerated coarsened Ag₃Sn particles. Comparing Fig. 4a, d, e, note that the large (~2μm) accelerated coarsened Ag₃Sn particles are often located at recrystallised grain boundaries and the interior of recrystallised grains are, in many places, particle free zones. In contrast, subgrains (e.g. blue regions in (c) and regions within blue boundaries in (d)) contain numerous Ag₃Sn particles that are only slightly coarser than those at the centre of the joint (e.g. see the top left region in Fig. 4d, e). Comparing (f) and (d), the regions with anomalously wide Ag₃Sn particle spacing (yellow triangles in (f)) match well with the regions of recrystallised grains in (d). A crack has propagated from the top-right along some of the grain boundaries between the recrystallised grains and the parent grain, Fig. 4a.

The features of microstructure evolution in Fig. 4 are similar to a large body of past work[12,22,39–42] including by Bieler et al.[1,41] and Arfaei et al.[11,12,22]. However, the extent and detail of these features after 7580 cycles in the 84CTBGA varied strongly from joint to joint depending on the initial microstructure as well as the location in the array. To enable a quantitative comparison across all studied joints, we define two measures of damage as overviewed in the example in Fig. 5a–d. First, we define the "cracking extent" as the ratio of the projected crack length to the top interface length in 2D sections (Fig. 5a). Second, as a measure of microstructural damage, we define the area fraction of the cross-section with MO higher than 8°. The rationale for selecting MO > 8° can be seen by comparing Fig. 4h with the rest of Fig. 4. In Fig. 4h, red has been assigned to regions with MO > 15°, yellow to regions with 8° ≤ MO ≤ 15°, and blue to regions with MO < 8°. The region with MO > 15° correlates well with the regions of recrystallisation and accelerated Ag₃Sn particle coarsening whereas the region with 8° ≤ MO ≤ 15° captures subgrains and regions of strong localised lattice curvature. The selection of a minimum value of 8° was to exclude small misorientations that are present even in time-zero joints (Fig. 2). The MO > 8° area is preferred here because lattice curvature and subgrain development are considered important aspects of damage accumulation in addition to recrystallisation. However, the conclusions of this paper are not significantly altered if MO > 15° is used. When calculating the area fraction with MO > 8°, we consider only the top quarter of the joint (see Fig. 5d) since this is where cracking and failure occurred. However, again, the results of this paper are not significantly altered if the whole joint area is used rather than the top quarter only.

Figure 5e plots the extent of cracking vs. the area fraction with MO > 8° for the 80 joints investigated in one thermally cycled 84CTBGA package. There is a high degree of scatter and a poor correlation. We note that measuring the extent of partial cracking in 2D (X-Y) cross-sections has limitations. In prior work[43], top-down (X-Z) sectioning of BGA packages has shown the complex shape of cracks as well as crack growth from multiple locations around the top circumference of the joint. It was demonstrated that the appearance of a partial crack in a 2D (X-Y) section would depend strongly on the location of the cross-section.

To proceed, we averaged the data for individual joints at equivalent locations in the package exploiting the 1/8 symmetry of the 84CTBGA as indicated in the insert of Fig. 5f. The correlation between the average extent of cracking and the area fraction with MO > 8° is

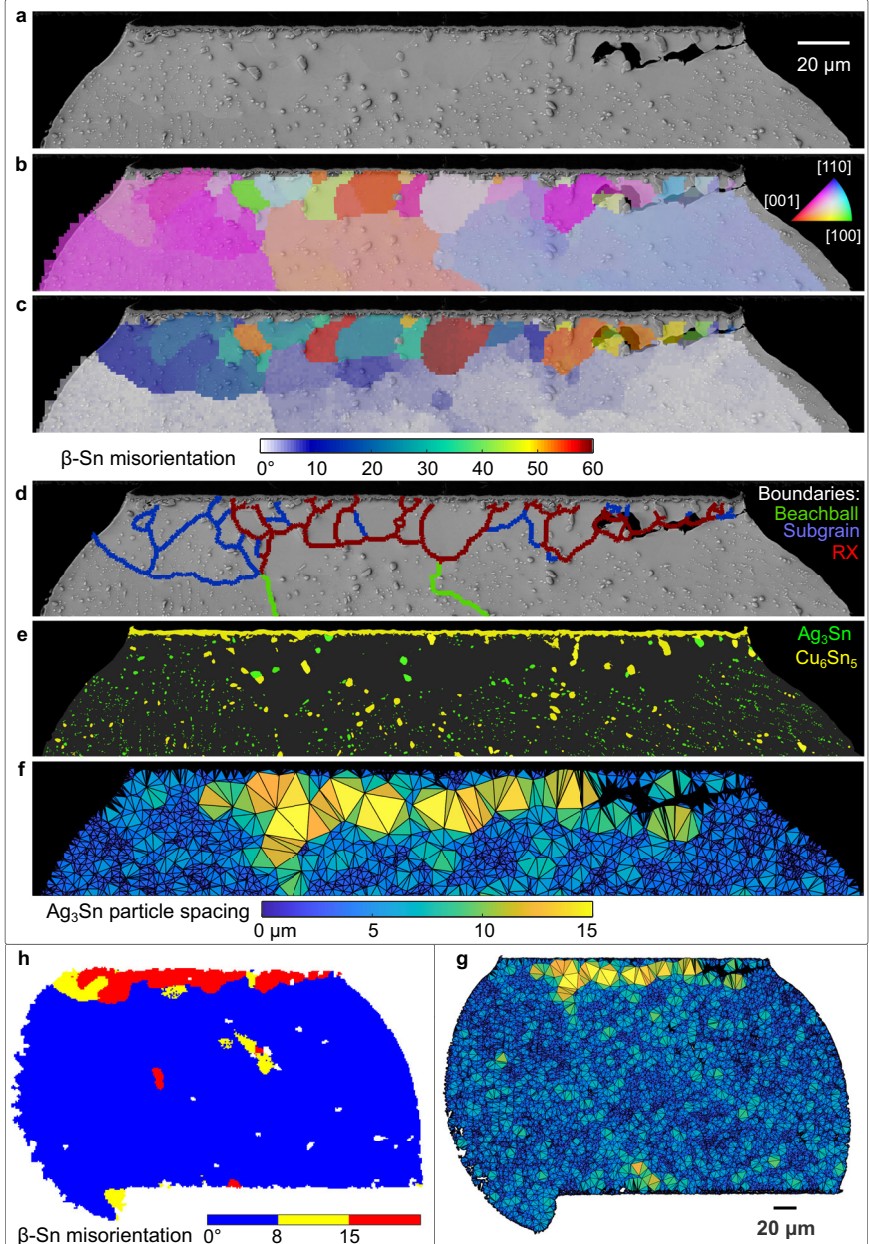

**Fig. 4 | Typical features of microstructure and damage evolution due to thermal cycling using joint G08 as an example. a** BSE-SEM image near the top of the joint, **b** EBSD orientation map with IPF-X colouring, **c** misorientation map with reference to the as-solidified major Sn orientations, **d** grain boundary segments categorised by misorientation: 5–15° = subgrains; >15° = recrystallised grain boundaries, **e** IMC layer and particle segmentation, **f, g** Ag$_3$Sn particle spacing maps generated with a Delaunay triangulation of all measured Ag$_3$Sn particle centroids. **h** Misorientation map using threshold ranges of 15° > MO > 8° and MO > 15°.

much stronger (Fig. 5f). We next selected individual joints for which the neutral point vector lies close to the sectioning plane as indicated in the insert of Fig. 5g. For these individual joints (Fig. 5g), the general correlation is much clearer than when considering all individual joints in Fig. 5e, albeit with some remaining scatter. Comparing Fig. 5e versus (g), the results are consistent with the extent of cracking depending on the cross-sectioning plane and indicate that the greatest error (in crack extent) occurs for joints where the neutral point direction is near-perpendicular to the cross-section (e.g. M6,7 and A6,7). It is likely that there are true microstructure effects that determine how the extent of cracking depends on the evolving microstructure and therefore MO; however, this could not be studied here due to the limitations of measuring partial cracks in 2D sections. The MO > 8° approach taken here is also a 2D method but is less prone to cross-sectioning effects since the MO features occupy a larger volume of the joint and EBSD

mapping captures more of the 3D information than the (essentially 1D) extent of cracking. Therefore, for the remainder of this paper, we consider MO > 8° as the measure of damage.

## Single grain joints

For all 15 single grain joints studied after thermal cycling, Fig. 6a plots the area fraction with MO > 8° versus the angle between the ⟨001⟩$_{Sn}$ and the Y-axis. There is a clear increase in damage as the angle increases, which is consistent with prior experimental studies that used other measures of damage[1–3]. The result in Fig. 6a can be partially understood from Fig. 6b which plots the mean CTE of β-Sn in the X-Z plane as a function of the angle between the ⟨001⟩$_{Sn}$ and the Y-axis. Also shown is the mean CTE of the package in the X-Z plane which was measured to be $11 \times 10^{-6}$ K$^{-1}$ in ref. 20. The mean in-plane CTE of β-Sn increases as the angle between the ⟨001⟩$_{Sn}$ and the Y-axis increases,

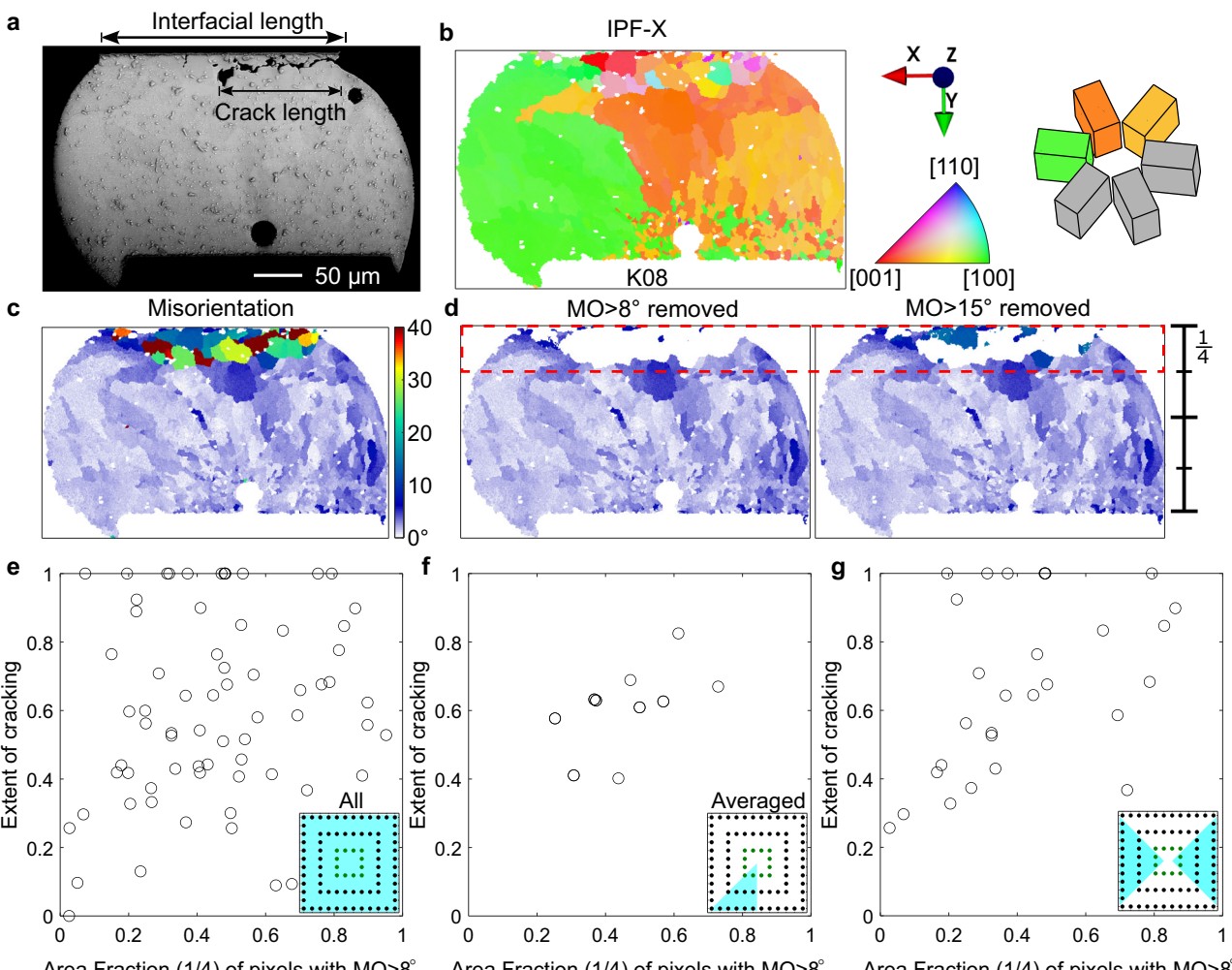

**Fig. 5 | Damage metrics. a** Definition of 2D cracking extent measured using a BSE-SEM image. **b–d** Definition of microstructural damage extent: (**b**) IPF-X map, (**c**) misorientation (MO) map, (**d**) filtered MO maps after removing MO > 8° and MO > 15°. **e** Plot of cracking extent vs. area fraction of pixels with MO > 8° from the top (1/4) region of all 80 investigated joints in a 84CTBGA. **f** The data from (**e**) after averaging symmetry-related positions in the array as indicated in the inserts in (**e**, **f**). **g** Data from individual joints in the cyan regions of the insert.

and the CTE mismatch with the package increases in a similar manner. However, this is only one factor. A more complete analysis needs to consider the CTE mismatch in other directions, the anisotropic elasticity and plasticity as well as the different boundary conditions at each position in the array of joints.

To do this, the multi-scale model was applied to the six single grain joints at locations A6, A7, A8, A9, M8, and M9 using the measured orientations for each joint. The distribution of stored energy density (SED) developed in three of the solder joints at the end of 10 thermal cycles is shown on the left of Fig. 6c. The localisation of SED occurs at hotspots along the top interface and particularly along the top circumference. Experimental MO maps in the X-Y cross-section are shown on the right of Fig. 6c, where regions of localised MO are in reasonable agreement with the simulated regions of high SED.

The peak in SED was found by averaging the highest 10% of SED values (i.e. the maximum 20 elemental values) along the top interface of each joint and these peak SED values are plotted against the angle between the $\langle 001 \rangle_{Sn}$ and the Y-axis in Fig. 6d. The experimental MO > 8° of the same six joints is given as a second axis. The correlation in Fig. 6d shows that the peak SED calculated with the CPFE model after 10 thermal cycles is a good indicator of the area fraction of microstructure occupied by subgrains and recrystallised grains after 7580 thermal cycles in the experiments.

## Multigrain joints

A key finding in this work is that many multigrain joints were more damaged than the worst-oriented single grain joints, both in terms of MO > 8° and in terms of cracking extent. EBSD analysis of five thermally cycled solder joints is shown in Fig. 7a, b, spanning the range of microstructure and damage in the CTBGA package. The text annotated above the unit cell wireframes summarises the number of cyclic twinned rings, whether nucleation occurred from the top or bottom side of the joint (deduced from the method in Fig. 3), and the number of original tin orientations contacting the top (package-side) of the joint where damage accumulates. Figure 7b are the corresponding generated misorientation maps with recrystalized grains in green and hotter colours. The regions with MO > 8° have been excluded from the EBSD IPF maps in Fig. 7a to emphasise the original solidification microstructure.

Examination of Figs. 7a, b and 2c, d reveals that, in these eight joints, the single grain joints developed less area fraction with MO > 8° than multigrain joints and, among the multigrain joints, the region with MO > 8° generally became more extensive as the number of original grains (orientations) at the top of the joint increased. This was also found to be the general trend for all thermally cycled joints in this study as plotted in Fig. 7c, which considers the 68 joints marked by filled circles in the pin diagram in Fig. 7a and splits the joints into three

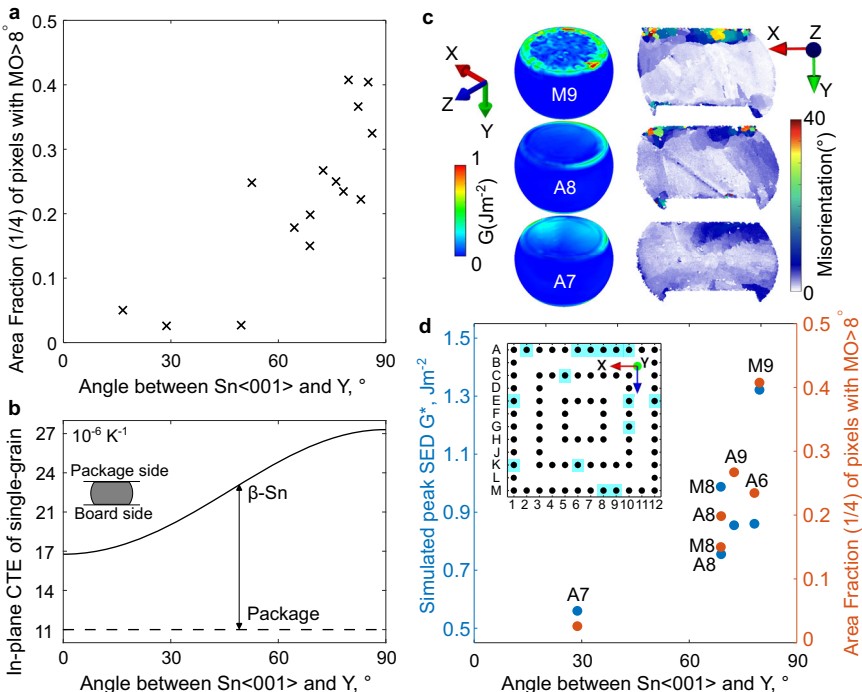

**Fig. 6 | Influence of tin grain orientation on the experimental damage and simulated stored energy density (SED) for single grain joints. a** Measured area fraction with MO > 8° versus the angle between Sn⟨001⟩ and Y for all 15 single grain joints studied. **b** Calculated connection between in-plane CTE and the angle between Sn⟨001⟩ and Y, compared with the in-plane CTE of the package. **c** Simulated 3D SED distribution and measured misorientation distribution in the 2D cross-section of three joints. **d** Correlation between simulated peak SED and experimental area fraction with MO > 8°.

groups by their microstructure: single grain; multigrain with a single original orientation at the top; and multigrain with multiple original orientations at the top. The innermost 12 joints were not included in Fig. 7c for clarity. Note in Fig. 7c that the most damaged single grain joints (which have the 'worst' orientations) were less damaged than some multigrain joints. This result is not consistent with statements in prior work that damage in multigrain joints depends on the mean orientation of grains[1,2,41].

To explore CTE mismatch in multigrain microstructures, Fig. 8 demonstrates connections between tin orientations measured by EBSD, the 3D CTE ellipsoids and the 2D section of the CTE in the plane of the substrate. Consider first the single grain solder joints in Fig. 8a, b. In Fig. 8a, the in-plane CTE of the β-Sn grain is the blue ellipse on a grey background and the dashed ellipse is the in-plane CTE of the 84CTBGA package from[20]. Figure 8b shows another (orange) single grain example with a larger substrate-plane CTE ellipse in the grey substrate plane because the ⟨001⟩Sn lies nearly parallel with the substrate.

Now consider beachball microstructures consisting of a ring of three twinned β-Sn orientations such as that in Fig. 8c, d. Figure 8e plots the triple CTE ellipsoids corresponding to the three cyclic twinned grains oriented as in Fig. 8c. Figure 8f then shows the digitally reorientated 3D triple ellipsoids with a translucent X-Z substrate plane, and its 2D section in the grey substrate plane plus the dashed ellipse of the in-plane CTE of the 84CTBGA package. This visualises and quantifies the in-plane CTE mismatch among the three β-Sn grains in the substrate plane and between these β-Sn grains and the package.

The CTE mismatch among the three tin orientations in the substrate plane depends on the orientation of the tricrystal with respect to the substrate. The same is true of double ring cyclic twins where there are five interrelated β-Sn orientations, and a similar procedure of 2D sectioning the centre of five ellipsoids in the substrate plane allows the in-plane CTE to be calculated. This is overviewed in Fig. 8g which plots the calculated maximum difference in the mean in-plane CTE (the max

ΔCTE) between β-Sn grains in the substrate plane (i.e. the difference in the mean CTE between the largest and smallest β-Sn ellipses on the grey substrate plane in Fig. 8f) for different single ring and double ring cyclic twinned β-Sn orientations. Our calculations assume that all β-Sn orientations contact the top package. The plots were generated by applying a set of 50,000 random orientations to single ring and double ring cyclic twins, calculating the maximum ΔCTE in the substrate plane for each, plotting the orientations as data points in Euler space, and colouring each data point by the calculated maximum ΔCTE between β-Sn grains in the substrate plane. The 50,000 maximum ΔCTE values are plotted as histograms in Fig. 8h for both single and double ring microstructure types, revealing various features: (i) there is overlap between the distributions such that some single ring microstructures have a higher maximum ΔCTE value than some double ring microstructures and vice versa; (ii) single ring orientations tend to result in milder maximum ΔCTE values; and (iii) double ring orientations have the potential to cause the most severe maximum ΔCTE values among tin grains in the substrate plane.

Using the approach in Fig. 8, the MO > 8° data in Fig. 7c are plotted against the CTE mismatch in Fig. 7d, e. For single grain joints and multigrain joints with a single orientation at the top, Fig. 7d plots the CTE mismatch between the package and the β-Sn grain contacting the top (package) side. This is the same CTE mismatch as the difference plotted in Fig. 6b. For single grain joints, the extent of damage generally increases with increasing in-plane ΔCTE to the package which is the same result as in Fig. 6a, although plotted as a CTE mismatch. For multigrain joints with a single orientation at the top in Fig. 7d, the MO > 8° data are slightly higher (although with a high degree of overlap) than the single grain data at the same CTE mismatch values.

For joints with two or more β-Sn orientations at the top, the MO > 8° data are significantly higher than for single grain joints in many cases. Figure 7e plots the extent of damage (area fraction with MO > 8°) versus the maximum ΔCTE between β-Sn grains. The data points are coloured by the number of β-Sn orientations at the top, e.g.

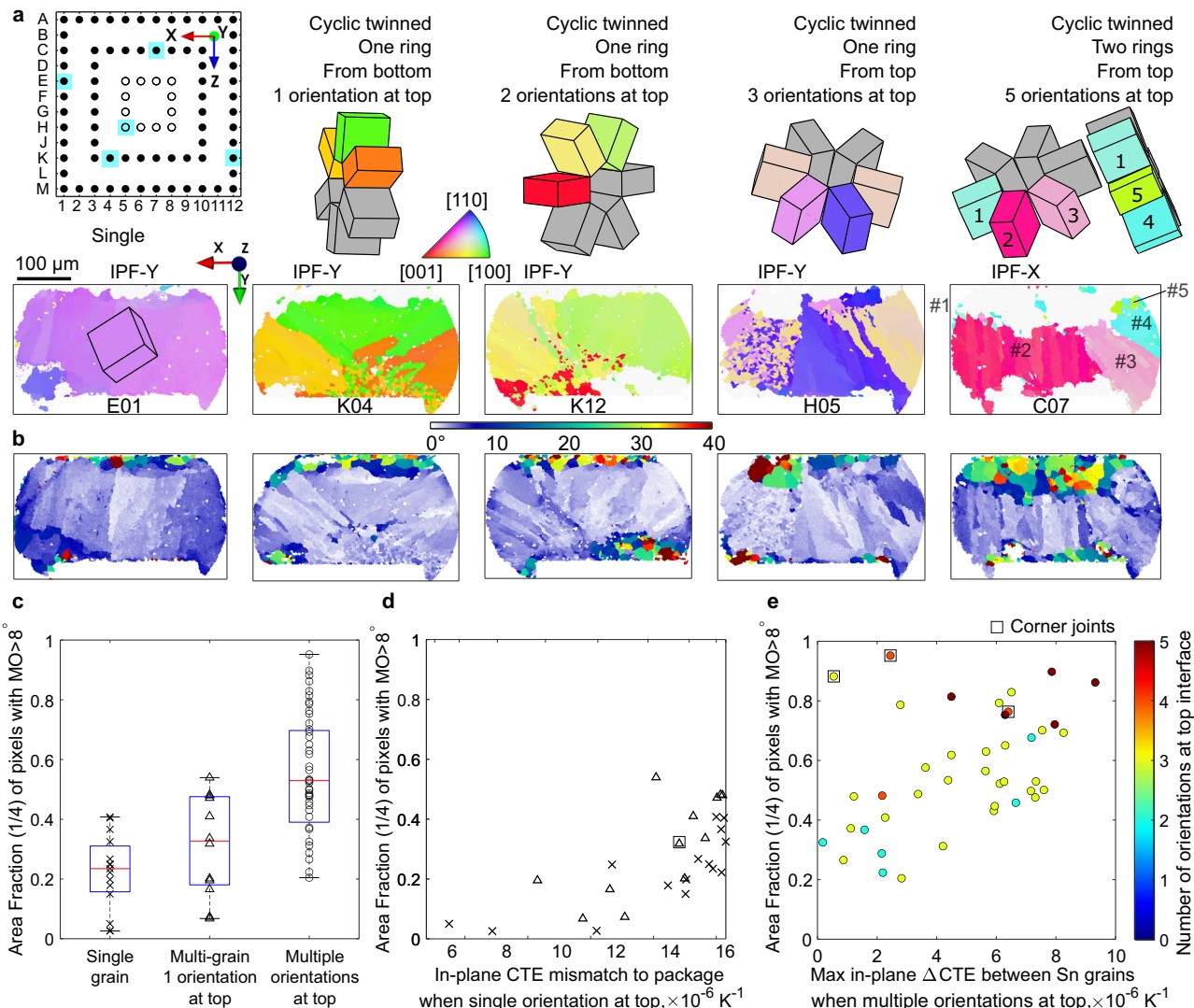

**Fig. 7 | EBSD and damage analysis of typical microstructures. a, b** Five examples spanning the range of microstructures observed after thermal cycling: (**a**) Sn IPF orientation maps with MO > 8° removed and measured unit cell wireframes, (**b**) misorientation maps defined with reference to the original major Sn orientations. **c** Damage area fraction values for different types of microstructure. **d** Area fraction with MO > 8° increases with in-plane CTE mismatch to package when single orientation at top (based on experimental 2D sections). **e** Area fraction with MO > 8° increases with increasing maximum in-plane ΔCTE between Sn grains when multiple original orientations at top.

the dark red points are joints with five β-Sn orientations (double ring cyclic twins) at the top. The extent of damage generally increases with increasing maximum in-plane ΔCTE between the β-Sn grains. We also see that joints with more β-Sn orientations at the top were more likely to have a high maximum in-plane ΔCTE, consistent with the probability distributions in Fig. 8g, h, and therefore accumulated more damage.

Comparing Fig. 7d, e, note that the vertical axis is the same and spans from 0 (no MO > 8° in the top quarter) to 1 (100% of the area in the top quarter having MO > 8°). The most damaged joints are those with both multiple β-Sn grains at the top and a high CTE mismatch between those grains. The least damaged joints are those with a single β-Sn grain at the top and a small CTE mismatch to the package. Note that Fig. 7d, e do not account for any effects of the location in the array (e.g. distance to the neutral point) except for excluding the inner ring, and yet there is a reasonable trend between the CTE mismatch created by the β-Sn orientations and the area fraction with MO > 8°. This highlights the strength of the β-Sn microstructure effect in determining damage accumulation in Sn-3Ag-0.5Cu solder joints in this package. At the same time, some location effects are evident. Outer corner

joints are marked with a square around their datapoint and two of the three corner joints in Fig. 7e are somewhat anomalous. It is possible that their high degree of damage resulted in minor as-solidified β-Sn grain orientations being missed and it is also likely that the different boundary conditions at these locations played a role.

## Multi-scale modelling on multi grain joints

To understand the effects of microstructure while accounting for the boundary conditions at different locations in the array, the multiscale model was applied to the joints at locations M09, C03, M07 and C07 as overviewed in Fig. 9. These joints contain a single β-Sn grain in M09, cyclic twinned β-Sn with a single ring in C03 and M07, and a cyclic twinned double ring in C07 (Fig. 9a). C03 grew from the bottom (PCB) side with apparently one β-Sn orientation at the top in the 2D section whereas M07 and C07 grew from the top side with three orientations at the top. The generated digital microstructures are shown in Fig. 9b, c, revealing that C03 has two β-Sn orientations at the top in 3D. The interlacing in C03, M07, and C07 could be inferred from the presence of some small cyclic twinned domains near the edge of the

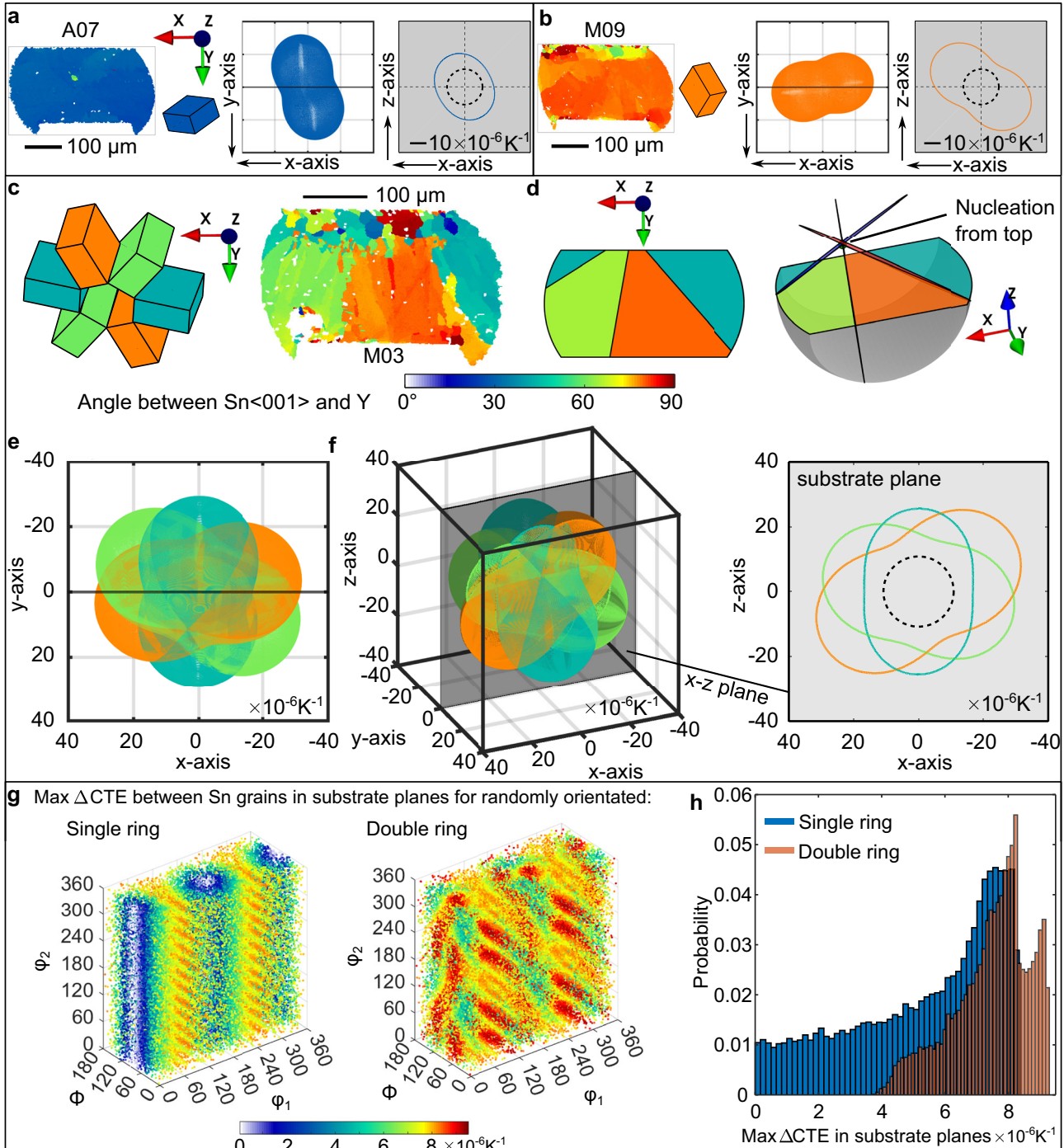

**Fig. 8 | Calculated connections between tin orientations and CTE mismatch.**
**a**, **b** Two single-grain examples: Sn IPF orientation map, unit cell, 3D CTE ellipsoid and 2D CTE in the substrate plane. **c** A beachball joint coloured by angle between Sn⟨001⟩ and Y plus cyclic Sn unit cells, **d** the geometric model at different viewing angles, **e** corresponding 3D CTE ellipsoids in the EBSD mapping coordinates, **f** 3D CTE triple ellipsoids at arbitrary viewing angle and 2D CTE section in the substrate

(x-z) plane. **g** Maximum in-plane CTE mismatch in the substrate plane between tin grains in single and double ring cyclic twins calculated from 50,000 random orientations and plotted in Euler space, **h** histograms of the max in-plane CTE mismatch from (**g**). On all in-plane 2D CTE plots (with grey background), the dashed ellipse is the in-plane CTE of the package.

recrystallised grains in the EBSD maps although the extent of the initial interlacing could not be determined accurately.

Geometrically necessary dislocation (GND) distributions were calculated and used to derive simulated MO distributions (based on the methodology in ref. 3,44) at the end of 10 thermal cycles in Fig. 9e. Experimental MO distributions are given in Fig. 9f. A different colour range is used since the experimental MO is after 7580 thermal cycles

when significant recrystallisation has occurred whereas the simulated MO is after 10 thermal cycles. We see that regions with high simulated MO in Fig. 9e correspond reasonably well with experimental regions of subgrains and recrystallisation in Fig. 9f. For example, in single grain joint M09, both the simulated MO hotspots and the measured regions of highest MO occur in a band adjacent to the top interface. In the multi-grain joints, the simulations contain localised MO within

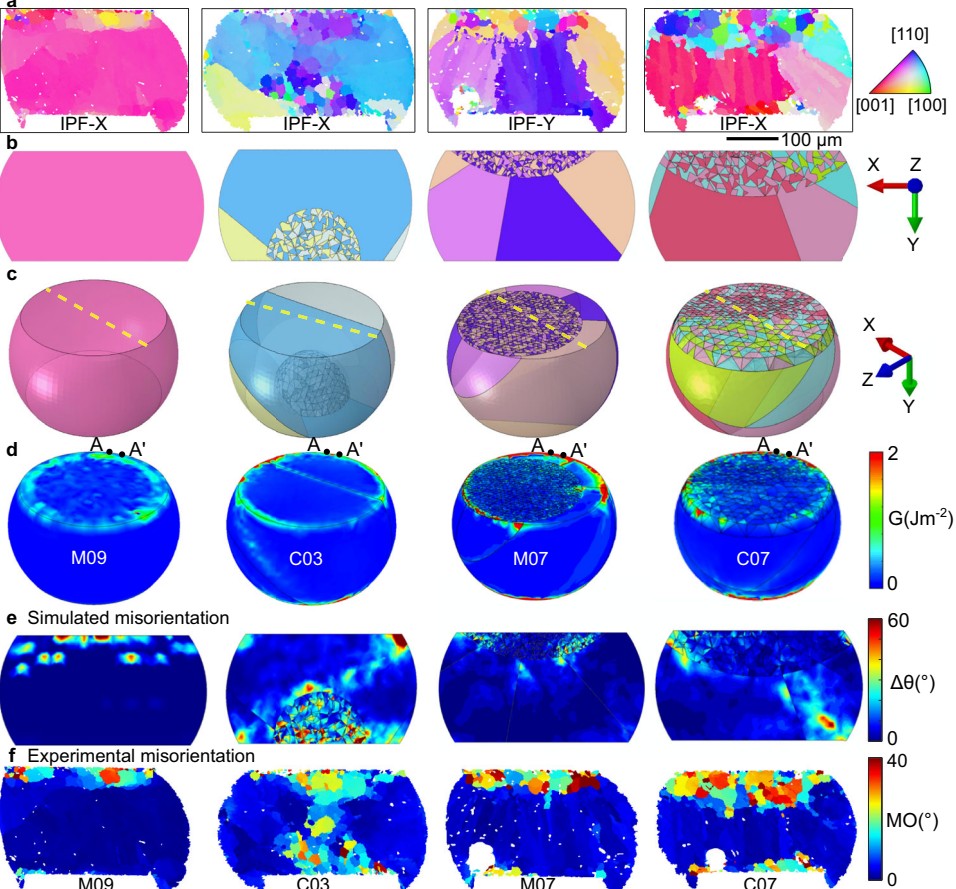

**Fig. 9 | Direct comparison between experimentally measured microstructure and damage and the multi-scale thermal cycling model of the same joints.** **a** Experimental EBSD IPF orientation maps after thermal cycling. **b**, **c** 3D geometrical models before thermal cycling, input into the CPFE model. Yellow dashed lines in (**c**) are the sectioned planes in (**b**) which are at the same location as in (**a**). **d** Simulated stored energy density distribution after 10 cycles. **e** Simulated misorientation distribution after the 10 cycles. **f** Experimental misorientation distribution after 7580 thermal cycles. **a**, **f** are experimental results. **b**–**e** are CPFE modelling.

interlaced regions which is also apparent in the experiments. This is particularly clear in C03 where the initial interlacing was on the bottom side and the simulated MO in this region reasonably matches the experimentally measured extent of high MO spreading from the bottom towards the centre of the joint. In the three multi-grain joints, there is a simulated MO hotspot at one or both top corners and the experimentally high MO spreads laterally from these regions adjacent to the top interface. In C07, the simulated MO contains an MO hotspot towards the bottom right of the joint and the experimental MO includes a region of high MO on the bottom side, slightly to the left of the simulated hotspot.

The predicted SED developed within these solder joints is shown in Fig. 9d at the end of 10 thermal cycles. Compared with the single grain joint M09, the SED in the multi-grain joints is more strongly localised at hotspots (peaks) along the top circumference in C03, M07 and C07, at sites associated with the intersections between βSn-βSn grain boundaries where stress concentrates[45]. To investigate the peaks in SED, the perimeter distribution of the SED along the path A-A′, indicated in Fig. 9d, is compared among the four joints in Fig. 10a. Note that the maximum on the SED colour scale in Fig. 9d is $G = 2\,\mathrm{J\,m^{-2}}$ whereas the peaks in SED in Fig. 10a are significantly higher. Thus, the hotspots in Fig. 9d are useful for confirming that peaks are located at the intersections of grain boundaries but cannot be used to compare the magnitude of the peaks. In Fig. 10a, we see that the peak SED developed in joint C07 is considerably larger than peaks in M07 and C03, consistent with the experimental damage observations in

Fig. 9a, f. Also, notice in Fig. 10a that the worst-oriented single grain joint (M09, see Fig. 6) develops relatively low peaks in SED compared with the multi-grain solder joints, consistent with the experimentally measured damage in Figs. 9 and 7d, e.

## Discussion

While the CPFE approach does not explicitly capture subgrain formation, recrystallisation and crack initiation, all these processes are directly driven by peaks in stored energy density (SED). For example, subgrains arise from thermally activated dislocation rearrangement and first develop in regions of locally high SED. During continued cycling, the same regions produce more dislocations that migrate to nearby subgrain boundaries, increase the subgrain boundary angle, and ultimately lead to continuous recrystallisation[41]. Similarly, a locally high SED is the driving force required to nucleate discontinuous recrystallisation[46,47] as well as to initiate a fatigue crack. Indeed, a similar SED criterion has been shown to predict fatigue crack nucleation sites in other ductile alloys including steels[48], nickel[49,50] and zirconium alloys[51].

Quantitative comparison between the peak SED predicted with the CPFE model and the experimentally measured damage is shown in Fig. 10b for the six single grain joints and three multigrain joints. The vertical axis is the average SED in the highest 10% values (i.e. the maximum 20 elemental values) along the top interface of each joint. Figs. 10b and 6d plot the same simulated SED data for the six single grain joints but with different SED axis limits. In Fig. 10b, all joints show

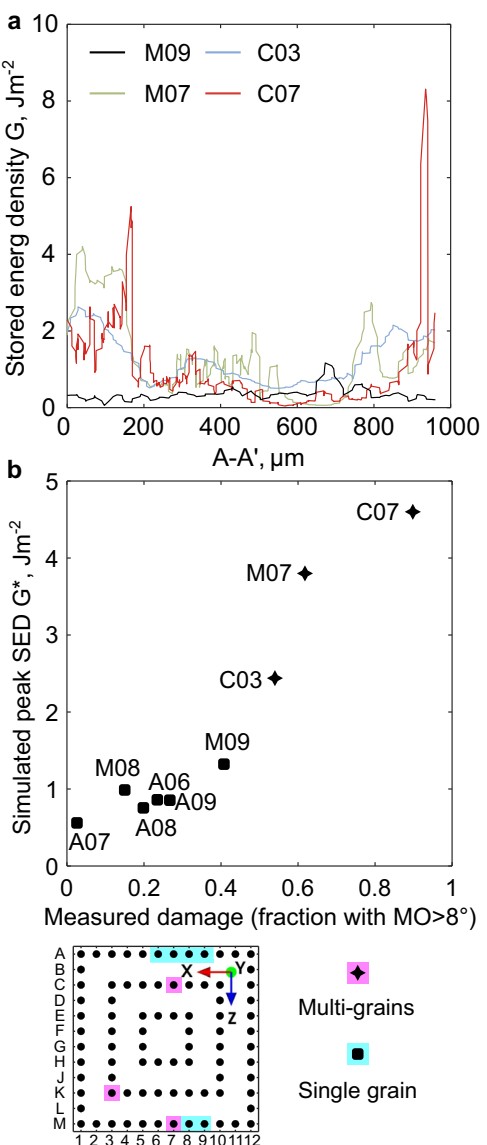

**Fig. 10 | Calculated stored energy in selected joints. a** Plots of stored energy density G along the perimeter of the top interface, i.e. path A–A' in Fig. 9d. **b** Representative stored energy G* that is averaged within the top 10% range around the peak positions in (**a**).

a strong correlation between the measured damage (area fraction with MO > 8°) and the simulated peaks in SED, consistent with local peaks in SED being the driving force for recrystallisation[47,52,53] and fatigue crack nucleation[48–50]. Comparing the simulations and experiments in Figs. 6, 9, and 10 highlights the key role of the magnitude of local peaks in SED in determining the extent of damage accumulation (area fraction of subgrains and recrystallisation in the region where cracking occurs) in this study.

In Figs. 9 and 10, we see that multi-grain joints develop higher peaks in SED than single grain joints due to stress concentration near grain boundaries where there is mismatch in CTE and elasticity due to the anisotropy of β-Sn. The negative effects of grain boundaries are most significant when the boundaries intersect the top interface where they accumulate the highest SED and when there is a large intrinsic CTE mismatch between these grains. Experimentally, in single grain joints, the MO > 8° increased as the angle between $\langle 001 \rangle_{Sn}$ and the Y-axis increased. Increasing this angle increases the in-plane mismatch in coefficient of thermal expansion (CTE) between the tin grain and the package (Fig. 7d). Many multigrain joints had significantly higher

damage (MO > 8°) than the worst-oriented single grain joints because there is additional CTE mismatch between grains which causes SED localisation near grain boundaries. The extent of damage increased as the CTE mismatch between grains in the substate plane increased (Fig. 7e).

The findings of this study reinforce the importance of the solder microstructure in determining the development of damage in solder joints. Our findings suggest that the optimum microstructure for thermal fatigue resistance is a single Sn grain with $\langle 001 \rangle_{Sn}$ perpendicular to the substrate. In multi-grain joints, avoiding grain boundaries intersecting the top interface and minimising the CTE mismatch between tin grains may also be useful. For our findings to be usable by industry, it will be necessary to develop industrial scale approaches to control the microstructure formed during the assembly/soldering process and combine this with alloy development. Some methods have been published to control tin orientations[24,54,55] or promote the formation of single grain joints[24] in Sn-Ag-Cu alloys, and certain alloying additions are also known to promote single grain joints[56]. It will be interesting in future work to optimise these microstructure control methods and measure the extent to which these approaches can improve the thermal fatigue performance of BGA and other packages.

## Methods
### Electronic test vehicle and thermal cycling
The test vehicle involved daisy-chained 84 input/output (IO) thin core chip array (84CTBGA) packages with an electrolytic Ni/Au pad finish. The printed circuit board (PCB) had high-temperature Panasonic R-1755V laminate and a surface finish of organic solderability preservative (OSP), with 16 sites for the 84CTBGA. The 84CTBGA packages had 300 μm Sn-3Ag-0.5Cu (wt%) balls attached and were assembled on the PCB using Sn-3Ag-0.5Cu (wt%) type 4 no-clean solder paste applied with a 130 μm thick stencil with 305 ×305 μm square apertures. The test vehicles were then reflowed in a 14-temperature zone convection oven in a $N_2$ atmosphere with a nominal peak temperature of 240 °C measured on the board adjacent to the solder joints. The test vehicle, assembly and soldering procedures are the same as ref. 20. A photograph of an assembled package is given in Fig. 1a. A pin diagram of the 84CTBGA package is shown in Fig. 2.

Two PCBs with a total of 32 84CTBGA packages were thermally cycled in accordance with the IPC-9701A guidelines[57]. The thermal profile involved ramping from 0/100 °C at 10°C per minute with 10-minute dwells at the hot and cold extremes of the cycle. The solder joints were monitored in situ using an event detector set at a resistance limit of 1000 ohms. The characteristic life ($N_{63.2}$) from a 2-parameter Weibull analysis of 32 packages in this thermal cycling profile was 7286 cycles. The strategy in this work was to apply a fixed number of thermal cycles to all joints in selected BGA packages and then to study how the accumulated damage depends on the solder microstructure and the location in the array. The thermally cycled packages studied here had undergone 7580 cycles.

### Microstructure characterisation
20 joints were studied in the as-soldered (time-zero) condition and 84 other joints were studied after thermal cycling. One thermally cycled package was given a near-complete analysis in which 80 out of 84 joints were characterised to study the combined effects of microstructure and location in the array on damage evolution. This involved grinding into the first row of joints, characterising the microstructures of each joint, grinding into the next row of joints, and so on through the package.

Packages were mounted vertically on the side of row M in Struers VersoCit-2 acrylic cold-mounting resin, ground down to 4000 grit SiC paper before being polished with OPS on a MD/DP-Nap pad. Back-scattered electron (BSE) images were taken using a Zeiss Sigma300 field emission gun scanning electron microscope (FEG-SEM). To

measure the eutectic $Ag_3Sn$ and $Cu_6Sn_5$, a fine imaging pixel size of 19.49 nm was used for time-zero samples and 38.98 nm for thermally cycled samples. To capture the entire joint at this resolution, the SEM stage was automated to scan across the joint, collecting $14 \times 10$ images for each time-zero joint and $7 \times 5$ images for each thermally cycled joint. The collected images were then stitched into a single image in ImageJ. Fine $Ag_3Sn$ and $Cu_6Sn_5$ particles were segmented using the in-house MATLAB image processing algorithms published in[36]. The eutectic $Ag_3Sn$ particle spacing was found from a Delaunay triangulation in MATLAB, using the centroids of all detected $Ag_3Sn$ particles as the vertices of the triangulation. Particle spacing maps were generated by filling the triangles with a colour scale that was proportional to the mean side length of each triangle.

Crystal orientation maps were collected by a Bruker e-Flash electron backscatter diffraction (EBSD) detector with the ESPRIT 2.1 software. All EBSD datasets were post-analysed using the MTEX toolbox in MATLAB. A data-processing pipeline was developed to characterise the misorientation maps for different types of β-Sn microstructures: single grain, cyclic twinned beachball and interlaced. For single grain joints, the reference orientation was taken as the orientation with the highest density in the MTEX orientation distribution function (ODF) near the centre of the joint and the misorientation of each pixel was calculated relative to this reference orientation. For joints with multiple β-Sn grains, the pipeline included: 1) select all major β-Sn orientations from the ODF near the joint centre as the reference orientations, 2) generate one misorientation map for each reference β-Sn orientation, 3) take the minimum misorientation value at each pixel and form a new map. The method developed here enabled misorientation maps to be quantified in complicated β-Sn microstructures and allowed numerous joints to be analysed within a single framework.

### Building 3D microstructure models from 2D EBSD maps

3D beachball microstructures were generated from 2D EBSD maps by measuring the crystallographic orientations of the β-Sn grains (from the ODF near the joint centre) and the location of the grain boundary vectors in the 2D section and using the following 3D information/assumptions. (i) A beachball is a cyclic twin with six twin segments made up of three β-Sn orientations related through 60° rotations around a common ⟨010⟩ axis[21]. (ii) The macroscopic grain boundaries are {101} planes (Fig. 3). (iii) The β-Sn nucleation point is on the $Cu_6Sn_5$ layer, either the top or the bottom layer[23]. (iv) Measured cross-sections such as Fig. 3 are vertical (X-Y) sections through the centre of the joint. An example of this construction is given in Fig. 3d. For joints with interlacing, an interlaced volume was generated by intersecting the joint volume with a spherical interlaced volume whose radius and location were tuned to reproduce a digital cross section similar to the EBSD cross section. The interlaced volume was then filled by Voronoi tetrahedra with sizes mimicking those in the EBSD map, and with the measured orientations randomly assigned among the interlaced grains.

### Multi-scale modelling of thermal cycling

Figure 1 overviews the parallel experiments and modelling approach. The multi-scale model couples (i) a macro-scale continuum model of the BGA package, solder joints and printed circuit board which considers the solder as an isotropic viscoplastic continuum with (ii) a dislocation based, temperature-dependant CPFE model of the individual solder joints, which accounts for the anisotropic thermal expansion, elasticity and plasticity of β-Sn and the embedded $Ag_3Sn$ and $Cu_6Sn_5$ intermetallics. These models are coupled by passing the thermomechanical transient predictions for the displacement of respective joints in the ball grid array from the macro board-level model to the microstructure-level model of an individual solder joint, thus defining the boundary conditions for the latter model.

### Macro-scale modelling at the package/board scale

The properties used for the individual materials in the PCB and package excluding the solder alloy are given in[3] (Table 1 in that paper). For the Sn-3.0Ag-0.5Cu material model, two different macro-scale continuum models were compared to investigate the effect of the model assumptions on the cyclic inelastic deformation (displacements) of solder joints. Approach 1 combined separate models for kinematic hardening plasticity (Chaboche formulation) and for creep (Garofalo-Arrhenius formulation). Approach 2 used the Anand model where the sum of plastic strain and creep strain are considered in a unified inelastic strain.

In approach 1, the non-linear kinematic hardening rule allows the modelling of cyclic hardening and can capture the Bauschinger effect. The kinematic hardening model of Sn-3.0Ag-0.5Cu deployed in this work is a rate-independent version of the kinematic hardening model for the back-stress tensor $\alpha$ proposed by Chaboche[58,59] and obtained by superimposing three evolving kinematic back-stress tensors $\alpha_i, i = 1,2,3$.

$$\alpha = \sum_{i=1}^{n} \alpha_i \qquad (1)$$

The evolution of each back-stress model is defined with the kinematic hardening rule,

$$\dot{\alpha}_i = \frac{2}{3} C_i \dot{\varepsilon}^{pl} - \gamma_i \dot{\bar{\varepsilon}}^{pl} \alpha_i \qquad (2)$$

where $C_i$ and $\gamma_i$ are model material parameters, $\dot{\varepsilon}^{pl}$ is the plastic strain rate, and $\dot{\bar{\varepsilon}}^{pl}$ is the magnitude of the plastic strain rate. The Chaboche model parameters for Sn-3.0Ag-0.5Cu are detailed in Supplementary Table 2.

The creep behaviour part of approach 1 was modelled with the Garofalo-Arrhenius constitutive law,

$$\dot{\varepsilon}^{cr} = A \left[ \sinh(B\sigma) \right]^n \exp\left( -\frac{Q}{RT} \right) \qquad (3)$$

where $\dot{\varepsilon}^{cr}$ is the steady-state creep strain rate, $\sigma$ is the applied stress, $T$ is the temperature in Kelvin, $A$ and $B$ are material constants, $R$ is the gas constant, $Q$ is an activation energy, and $n$ is a stress exponent. The Garofalo-Arrhenius creep model reported in[60] was deployed with temperature-dependant values of the material constant $A$ to capture the creep behaviour of Sn-3.0Ag-0.5Cu. The creep constitutive law parameters are detailed in Supplementary Table 3.

For approach 2, the Anand model constitutive law was used for the inelastic strain rate of Sn-3.0Ag-0.5Cu and the respective model constants were the same as those reported in[3] (Table 2 in that paper).

The finite element analysis (FEA) revealed that the displacement predictions for the solder joints in approach 1 (separate kinematic hardening plasticity and pure creep constitutive laws) were very closely matched to the equivalent simulation run with approach 2 (the Anand model with unified inelastic strain rate). Supplementary Fig. 2 and Supplementary Fig. 3 show a comparative analysis of the displacement results for solder joint M09 from the two macro-scale model approaches, showing that nodal displacements calculated from the two approaches are very similar, and the extreme point displacements in the solder joint differ by at most 3.7% and generally by <1%. Statistical analysis of the relative difference in predicted displacements for all solder joint mesh nodes, predicted with the two different material constitutive law approaches for Sn-3Ag-0.5Cu, show that the mean (average) value of the difference is only 0.8% and the respective standard deviation is 0.9%. This high similarity suggests that, in this study, kinematic hardening does not have a significant effect on the deformation of solder joints, and that displacement is dominated by

creep. Several reasons for this are identified: (1) Sn-3.0Ag-0.5Cu has a high homologous temperature over the investigated mild temperature cycle 0 °C to 100 °C ($T/T_m = 0.54 – 0.74$) which makes creep behaviour strongly dominant over kinematic hardening plasticity, (2) the long temperature cycle duration of 40 minutes, with 10-minute dwells at both the low (0 °C) and the high (100 °C) temperature extreme, is associated with low strain rates that favour creep deformation and stress relaxation throughout the cycle, (3) solder joint movements at the component assembly level are kinematically constrained by the movement of the printed circuit board and the BGA package. While these conclusions have been drawn for the problem in this paper, solder joints subjected to temperature cycles with lower-temperature regimes and fast cycling rates could promote kinematic hardening behaviour over creep. Due to the similar results from the two macro-scale model approaches here, only the Anand model transient displacement boundary conditions were passed over to the microstructure level CPFE model.

### Crystal plasticity finite element model at the microstructure scale

For the CPFE model, the 3D microstructure models (built from 2D EBSD maps) and the temperature-dependent, anisotropic material properties of Sn-3.0Ag-0.5Cu in Supplementary Table 4[61–64] were used. These material properties were extracted from a mechanics-based homogenisation of β-Sn and embedded IMC phases, where the material properties were obtained through a combination of crystal plasticity modelling and micro-pillar experimental tests[61]. The homogenisation also incorporates the IMC size effect into the hardening rate.

In the CPFE model, the slip rate $\dot{\gamma}^{(i)}$ along the $i^{th}$ slip system is determined by temperature-dependent thermal activation events where pinned dislocations jump over obstacles (such as lattice defects, inclusions etc.) in both forward and backward directions, which is given by:

$$\dot{\gamma}^{(i)} = \rho_{ssdm}b^2 v_D \exp\left(-\frac{\Delta H}{k\theta}\right) \sinh\left(\frac{\Delta V}{k\theta}\left|\tau^{(i)} - \tau_c^{(i)}\right|\right) \qquad (4)$$

where $\rho_{ssdm}$ is the mobile statistically stored dislocation density, $b$ the magnitude of the Burgers vector, $v_D$ the dislocation jumping frequency, $\Delta H$ the activation energy, $\Delta V$ activation volume, $k$ the Boltzmann constant, and $\tau^{(i)}$ and $\tau_c^{(i)}$ are the resolved shear stress and critical resolved shear stress on the $i^{th}$ slip system, respectively.

The geometrically necessary dislocation (GND) density term $\rho_{GND}$ is established from the plastic strain gradients that accommodate lattice curvature, which are calculated from Nye's dislocation tensor $\boldsymbol{\Lambda}$ that originates from the curl of plastic deformation gradient $\boldsymbol{F^p}$:

$$\boldsymbol{\Lambda} = \boldsymbol{\nabla} \times \boldsymbol{F^p} \qquad (5)$$

A microstructure-sensitive, Griffith-Stroh type energy-density driving force is employed to investigate fatigue damage that is given by:

$$G = \int \frac{\xi \boldsymbol{\sigma} : \mathrm{d}\boldsymbol{\varepsilon}^p}{\sqrt{\rho_{ssd} + \rho_{gnd}}} \qquad (6)$$

where $G$ is the stored energy density (SED) and $\xi$ is the fraction of the total plastic energy stored in the dislocation structure, taken as 0.05[65]. $\rho_{ssd}$ is the statistically stored dislocation density and $\rho_{gnd}$ is the geometrically necessary dislocation density.

The statistically stored dislocation density accumulation rate is related to the effective plastic strain rate $\dot{p}$ and current accumulated plastic strain $p$ according to:

$$\dot{\rho}_{ssd} = \lambda_1 \dot{p} - \lambda_2 p \quad (\mu m^{-2} s^{-1}) \qquad (7)$$

where $\lambda_1$ and $\lambda_2$ control the slip system hardening and recovery rates respectively and depend on the average $Ag_3Sn$ and $Cu_6Sn_5$ IMC radius as given in Supplementary Table 4.

## Data availability
Datasets including EBSD maps and SEM images are available from the corresponding author on request.

## Code availability
The custom algorithm/code to calculate the damage metric is a simple data-processing algorithm and explained in the "Methods" section "Microstructure characterisation". The algorithm used for image segmentation was previously published and cited in the main text. The macro-scale models use standard finite element technology and commercial software tools/solvers. The codes or software related to micro-scale modelling are routine analysis methods. All code/algorithm can be provided upon request to the corresponding author.

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

## Acknowledgements
We thank Prof. Chris Bailey for his input early in this project. The authors gratefully acknowledge the use of characterisation facilities within the Harvey Flower Electron Microscopy Suite, Department of Materials, Imperial College London. This work was partially funded by UK EPSRC grants EP/R018863/1 (C.M.G.) and EP/R019207/1 (S.S.).

## Author contributions
R.J.C. produced the electronic test vehicles, conducted the thermal cycling, and contributed to the discussion; J.W.X. performed the materials characterisation, the data processing pipeline, and the CTE mismatch analysis and contributed to the discussion; S.S. performed and discussed the macro-scale FE analysis; Y.L.X. performed and discussed the CPFE analysis; C.M.G. and F.P.E.D. developed and supervised the project and contributed to the analysis and discussion. All authors wrote parts of the paper.

## Competing interests
The authors declare no competing interests.
