## [Peer Review File · Nature Communications]

The role of microstructure in the thermal fatigue of solder jointsREVIEWER COMMENTS

Reviewer #1 (Remarks to the Author):

The manuscript contains interesting and valuable information; the results are convincing and definitely useful for the sector using electronics in reliability-critical systems. However, this reviewer has the following feedback on the structure of the manuscript.

The manuscript contains many references, which, in this reviewer's opinion, is too many (83 references). Sometimes, more than 10 references are cited as a range by the authors in the text (for example, "...reported widely for SAC305 27,**29-41**...") without giving any acknowledgement to the authors of the cited papers.

The manuscript has many links to other (the authors' own) papers, both in the results and the description of the methodology, making reading and interpretation difficult. The reader must gather the methodology details from about 5-6 cited papers. Yet, for example, the sample preparation cannot be considered reproducible since even the cited articles lack essential information. For example, the soldering profile and its characteristics (particularly the Q_{η} heating factor and the cooling rate), which affect the structure of the BGA solder joints, are unknown. In chapter 3.2, the authors mention that "some joints" were studied (both for zero-time and after thermal cycling), but how much was this exactly?

The manuscript is sometimes not sufficiently focused. In the results, for example, subsection 2.2 (Microstructure evolution...) starts with an almost full page of literature review-like introduction (discussion?), and only then do the authors present the results. Here and in other chapters, it is advantageous to present the results first, and then discuss them, supported by the citations. The comparison of the two main types of solder joint grain structures (single-grain and multi-grain) is only indirectly done; the authors discuss the relevant results in two separate subsections (2.3. and 2.5). It would be worthwhile to compare the results directly.

This reviewer suggests that the authors limit the number of references used, provide details of the methodology for both sample preparation and modelling (if the cited articles do not include that detail), condense the manuscript, and emphasise the relevant points and novelties more.

Reviewer #2 (Remarks to the Author):

The manuscript entitled "Microstructure effects on the thermal fatigue of solder joints: 2 coupling damage measurements with multi-scale modelling" has been investigated in detail.

The topic addressed in the manuscript is potentially interesting and the manuscript contains some practical meanings, however, there are some issues which should be addressed and highlighted by the authors:

1. the paper contains originality and significant information adequate to justify publication.
2. Acceptable literature review, the paper seems to understand the research area that it covers. However, it is good if the authors add more recent references.
3. Acceptable methodology. This paper used published methodology according to some references.
4. The author shall explain why only observed the time-zero and after thermal cycle samples. What happens to the variation of the thermal cycle?
5. This is then applied to 19 measure the damage accumulated in 20 time-zero and 84 thermally cycled joints from 84CTBGA 20 packages. Please rewrite this sentence. Quite hard to understand.
6. ODF-please put the full term before the abbreviation at the page 5
7. Why is the methods in the section 3 and the result in the section 2. Is it according to the journal

format?

8. The results presented clearly and analysed appropriately. This paper combined the experimental and modelling approaches.
9. The results of performed investigations can be usable in the industry if it is well presented in term of fundamental knowledge of the microstructure (grain arrangement, misorientation etc) towards thermal fatigue damage.
10. The paper presentation such as sentence structure, jargon use, acronyms could be improved for better readability.
11. The authors should clearly emphasize the contribution of the study. Please note that the up to date of references will contribute to the up to date of your manuscript.
12. References are adequate. However, need to be put more recent references 2020-2024.

Reviewer #3 (Remarks to the Author):

The attempt to correlate the information of strain distribution in solder joints subjected to cyclic thermal loading estimated by conducting multi-scale FEA and the EBSD analysis results of thermally damaged solder joints is exciting. It can provide valuable knowledge for optimizing solder microstructures.

However, there is a great concern about the accuracy of multi-scale FEA because the Anand model was used in the macro-scale FEA for predicting the deformation behavior of solder even though this paper discusses fatigue damage of solder joints due to cyclic thermal loading. The FEA for predicting cyclic inelastic deformation must employ a constitutive model based on the kinematic hardening rule to express the Bauschinger effect. However, the Anand model is a model based on the isotropic hardening rule and it cannot predict cyclic plasticity well. Then, the boundary conditions derived from the macro-scale FEA using the Anand model are not suitable for use in the micro-scale FEA of solder joints.

The reviewer strongly recommends that you conduct all the macro-scale FEA in this study again by employing the other constitutive model that can predict both the cyclic and the time and temperature-dependent deformations of solder accurately. Otherwise, the SED with the high accuracy necessary for discussing the mechanism of fatigue damage of solder joints will not be obtained.

Response to reviewers' comments:

We thank the Reviewers for their careful consideration and helpful suggestions. In this response, we have copied the Reviewer text in blue and written our responses in black. In our revised manuscript, we have highlighted the added/changed text with cyan highlighter.

Reviewer #1 (Remarks to the Author):

The manuscript contains interesting and valuable information; the results are convincing and definitely useful for the sector using electronics in reliability-critical systems.

Thank you.

However, this reviewer has the following feedback on the structure of the manuscript. The manuscript contains many references, which, in this reviewer's opinion, is too many (83 references). Sometimes, more than 10 references are cited as a range by the authors in the text (for example, "...reported widely for SAC305 27,29-41...") without giving any acknowledgement to the authors of the cited papers.

Thank you. We have carefully checked the references and reduced the number from 83 to 65. This has involved removing the large ranges of authors and focussing instead on both the first pioneering studies and more recent studies. We have also explicitly added the author names to the text from selected cited papers.

The manuscript has many links to other (the authors' own) papers, both in the results and the description of the methodology, making reading and interpretation difficult. The reader must gather the methodology details from about 5-6 cited papers. Yet, for example, the sample preparation cannot be considered reproducible since even the cited articles lack essential information. For example, the soldering profile and its characteristics (particularly the Q_{η} heating factor and the cooling rate), which affect the structure of the BGA solder joints, are unknown.

To address this comment, we have expanded the Methods section to provide more detail so that readers do not need to look through past papers. The new subsection "Electronic test vehicle and thermal cycling" now reads:

"The test vehicle involved daisy-chained 84 input/output (IO) thin core chip array (84CTBGA) packages with an electrolytic Ni/Au pad finish. The printed circuit board (PCB) had high temperature Panasonic R-1755V laminate and a surface finish of organic solderability preservative (OSP), with 16 sites for the 84CTBGA. The 84CTBGA packages had 300 μm Sn-3Ag-0.5Cu (wt%) balls attached and were assembled on the PCB using Sn-3Ag-0.5Cu (wt%) type 4 no-clean solder paste applied with a 130 μm thick stencil with 305 x 305 μm square apertures. The test vehicles were then reflowed in a 14-temperature zone convection oven in a N_2 atmosphere with a nominal peak temperature of 240 $^{\circ}\text{C}$ measured on the board adjacent to the solder joints. The test vehicle, assembly and soldering procedures are the same as those used in ref. ²⁰. A photograph of an assembled package is given in Figure 1(a). A pin diagram of the 84CTBGA package is shown in Figure 2.

Two PCBs with a total of 32 84CTBGA packages were thermally cycled in accordance with the IPC-9701A guidelines ⁵⁷. The thermal profile involved ramping from 0/100 $^{\circ}\text{C}$ at 10 $^{\circ}\text{C}$ per minute with 10-minute dwells at the hot and cold extremes of the cycle. The solder joints were monitored *in situ* using an event detector set at a resistance limit of 1000 ohms. The characteristic life ($N_{63.2}$) from a 2-parameter Weibull analysis of 32 packages in this thermal cycling profile was 7286 cycles. The strategy in this work was to apply a fixed number of thermal cycles to all joints in selected BGA

packages and then to study how the accumulated damage depends on the solder microstructure and the location in the array. The thermally cycled packages studied here had undergone 7580 cycles. ”

Regarding the comment that the sample preparation cannot be considered reproducible, we note that engineers can choose different assembly steps and all reflow ovens are different. Importantly, the wide range of BGA microstructures that solidified in this paper are similar to those reported by different groups across the literature, and our paper demonstrates and explains how these different microstructures affect thermal fatigue performance.

In chapter 3.2, the authors mention that "some joints" were studied (both for zero-time and after thermal cycling), but how much was this exactly?

Sorry for the lack of detail here. We have now changed this text into “20 joints were studied in the as-soldered (time-zero) condition and 84 other joints were studied after thermal cycling.”

The manuscript is sometimes not sufficiently focused. In the results, for example, subsection 2.2 (Microstructure evolution...) starts with an almost full page of literature review-like introduction (discussion?), and only then do the authors present the results. Here and in other chapters, it is advantageous to present the results first, and then discuss them, supported by the citations.

Thank you for this comment. We agree it improves the readability. To address this comment, in the ‘microstructure evolution and damage metrics’ subsection, we have moved our results to the beginning and then added a more concise and focussed comparison of our results with the literature, supported citations. This has significantly shortened this section which helps focus the manuscript.

The comparison of the two main types of solder joint grain structures (single-grain and multi-grain) is only indirectly done; the authors discuss the relevant results in two separate subsections (2.3. and 2.5). It would be worthwhile to compare the results directly.

Thank you. To compare single-grain and multi-grain joints directly, we have rearranged some of the text into a short new Discussion section. The relevant new part is:

“In Figures 9 and 10, we see that multi-grain joints develop higher peaks in SED than single grain joints due to stress concentration near grain boundaries where there is mismatch in CTE and elasticity due to the anisotropy of β -Sn. The negative effects of grain boundaries are most significant when the boundaries intersect the top interface where they accumulate the highest SED and when there is a large intrinsic CTE mismatch between these grains. Experimentally, in single grain joints, the $MO>8^\circ$ increased as the angle between $\langle 001 \rangle_{Sn}$ and the Y-axis increased. Increasing this angle increases the in-plane mismatch in coefficient of thermal expansion (CTE) between the tin grain and the package (Figure 7d). Many multigrain joints had significantly higher damage ($MO>8^\circ$) than the worst-oriented single grain joints because there is additional CTE mismatch between grains which causes SED localisation near grain boundaries. The extent of damage increased as the CTE mismatch between grains in the substrate plane increased (Figure 7e)”

This reviewer suggests that the authors limit the number of references used, provide details of the methodology for both sample preparation and modelling (if the cited articles do not include that detail), condense the manuscript, and emphasise the relevant points and novelties more.

In summary, to address these points, we have made the following changes (more detail was given with each point above).

- We have reduced the number of references to 65 and explicitly mentioned the names of authors in some places of the text.
- We have rewritten the 'Electronic test vehicle and thermal cycling' section of the Methods to provide further details so that readers do not need to look elsewhere.
- We have expanded the 'Macro-scale modelling at the package/board scale' section of the Methods to provide more detail and address the comments of Reviewer #3.
- We have condensed the manuscript through editing, rearranging the text and deleting much of the literature review-like introduction which appeared before some of our results in the original submission.
- To emphasise the relevant points and novelty, we have reworked the text to have a short, focussed Discussion section at the end of the manuscript.

Reviewer #2 (Remarks to the Author):

The manuscript entitled "Microstructure effects on the thermal fatigue of solder joints: coupling damage measurements with multi-scale modelling" has been investigated in detail.

The topic addressed in the manuscript is potentially interesting and the manuscript contains some practical meanings, however, there are some issues which should be addressed and highlighted by the authors:

1. the paper contains originality and significant information adequate to justify publication.

Thank you.

2. Acceptable literature review, the paper seems to understand the research area that it covers. However, it is good if the authors add more recent references.

Thank you. We have carefully updated the references. In the revised manuscript, 24 out of the 65 references are to papers published since the year 2020. The remaining references are to older original studies.

3. Acceptable methodology. This paper used published methodology according to some references.

Thank you.

4. The author shall explain why only observed the time-zero and after thermal cycle samples. What happens to the variation of the thermal cycle?

We did not study packages that had experienced different numbers of temperature cycles because our characterisation method is destructive and, if we examine different packages, the microstructure will be different at each location in the array, so the development of damage with cycle number cannot be studied meaningfully. To provide more detail on our strategy, we have added the

following text near the end of the 'Electronic test vehicle and thermal cycling' section of the Methods section:

"The strategy in this work was to apply a fixed number of thermal cycles to all joints in selected BGA packages and then to study how the accumulated damage depends on the solder microstructure and the location in the array."

5. This is then applied to 19 measure the damage accumulated in 20 time-zero and 84 thermally cycled joints from 84CTBGA 20 packages. Please rewrite this sentence. Quite hard to understand.

Sorry for the unclear sentence. We have now rewritten it in the following way: "This approach is then applied to measure the damage accumulated in 20 time-zero joints and 84 thermally cycled joints from an 84CTBGA (thin chip array BGA) electronic test vehicle."

6. ODF-please put the full term before the abbreviation at the page 5

Sorry for not defining this acronym. We have now added the following text to the Methods section: "orientation distribution function (ODF)" at the page 17.

7. Why is the methods in the section 3 and the result in the section 2. Is it according to the journal format?

Yes, this is the journal format. The Methods section comes at the end of the manuscript. To make this clear to readers, we have added the following sentence to the end of the Introduction: "Detailed methods are given at the end of the manuscript."

8. The results presented clearly and analysed appropriately. This paper combined the experimental and modelling approaches.

Thank you.

9. The results of performed investigations can be usable in the industry if it is well presented in term of fundamental knowledge of the microstructure (grain arrangement, misorientation etc) towards thermal fatigue damage.

Thank you. To make this clear, we have added the following text to the final paragraph of Discussion:

"For our findings to be usable by industry, it will be necessary to develop industrial scale approaches to control the microstructure formed during the assembly/soldering process and combine this with alloy development. Some methods have been published to control tin orientations^{24,54,55} or promote the formation of single grain joints²⁴ in Sn-Ag-Cu alloys, and certain alloying additions are also known to promote single grain joints⁵⁶. It will be interesting in future work to optimise these microstructure control methods and measure the extent to which these approaches can improve the thermal fatigue performance of BGA and other packages."

10. The paper presentation such as sentence structure, jargon use, acronyms could be improved for better readability.

We have carefully re-read the manuscript and edited the sentence structure for improved readability. To remove jargon, we have changed SAC305 into Sn-3Ag-0.5Cu throughout and explained the meaning of the term “beachball”. We have ensured that all acronyms are now defined on first use.

11. The authors should clearly emphasize the contribution of the study. Please note that the up to date of references will contribute to the up to date of your manuscript.

Thank you for this suggestion. We now have 24 references since 2020 in the revised manuscript. We have also added the following paragraph to the end of the Discussion section to emphasize the contribution of the study more clearly.

“The findings of this study reinforce the importance of the solder microstructure in determining the development of damage in solder joints. Our findings suggest that the optimum microstructure for thermal fatigue resistance is a single Sn grain with $\langle 001 \rangle_{\text{Sn}}$ perpendicular to the substrate. In multi-grain joints, avoiding grain boundaries intersecting the top interface and minimising the CTE mismatch between tin grains may also be useful.”

12. References are adequate. However, need to be put more recent references 2020-2024.

We now have 24 references since 2020 in the revised manuscript.

Reviewer #3 (Remarks to the Author):

The attempt to correlate the information of strain distribution in solder joints subjected to cyclic thermal loading estimated by conducting multi-scale FEA and the EBSD analysis results of thermally damaged solder joints is exciting. It can provide valuable knowledge for optimizing solder microstructures.

We are very pleased that Reviewer 3 finds our approach exciting.

However, there is a great concern about the accuracy of multi-scale FEA because the Anand model was used in the macro-scale FEA for predicting the deformation behavior of solder even though this paper discusses fatigue damage of solder joints due to cyclic thermal loading. The FEA for predicting cyclic inelastic deformation must employ a constitutive model based on the kinematic hardening rule to express the Bauschinger effect. However, the Anand model is a model based on the isotropic hardening rule and it cannot predict cyclic plasticity well. Then, the boundary conditions derived from the macro-scale FEA using the Anand model are not suitable for use in the micro-scale FEA of solder joints.

The reviewer strongly recommends that you conduct all the macro-scale FEA in this study again by employing the other constitutive model that can predict both the cyclic and the time and temperature-dependent deformations of solder accurately. Otherwise, the SED with the high accuracy necessary for discussing the mechanism of fatigue damage of solder joints will not be obtained.

Thank you for this important suggestion. We agree that kinematic hardening is important in fully reversed cyclic plasticity and it is necessary to assess its role in the thermal fatigue of solder joints.

To address this, we re-ran the macro-scale FEA again employing a different constitutive model that incorporates kinematic hardening. We then quantitatively compared the results from the new FEA with Kinematic hardening and creep behaviour for solder, and the original Anand FEA. Our analysis shows that the displacements (which are transferred from the macro-FE model to the micro-CPFE model) calculated from the two approaches are very similar, with an average value of the difference in solder nodal displacements of 0.8%, and differences of at most 3.7% and generally <1% in terms of displacement minimum and maximum extremes. This high similarity suggests that, in this study, kinematic hardening has only a small effect on the deformation of solder joints and our analysis provides new insights into why this is the case. Based on the new FEA, the following text has been added as a large subsection in the methods titled: "Macro-scale modelling at the package/board scale" which we feel significantly improves the paper.

"For the Sn-3.0Ag-0.5Cu material model, two different macro-scale continuum models were compared to investigate the effect of the model assumptions on the cyclic inelastic deformation (displacements) of solder joints. Approach 1 combined separate models for kinematic hardening plasticity (Chaboche formulation) and for creep (Garofalo-Arrhenius formulation). Approach 2 used the Anand model where the sum of plastic strain and creep strain are considered in a unified inelastic strain.

In approach 1, the non-linear kinematic hardening rule allows the modelling of cyclic hardening and can capture the Bauschinger effect. The nonlinear kinematic hardening model of Sn-3.0Ag-0.5Cu deployed in this work is a rate-independent version of the kinematic hardening model for the back-stress tensor α proposed by Chaboche^{58,59} and obtained by superimposing three evolving kinematic back-stress tensors α_i , $i = 1,2,3$.

$$\alpha = \sum_{i=1}^n \alpha_i \quad (1)$$

The evolution of each back-stress model is defined with the kinematic hardening rule,

$$\dot{\alpha}_i = \frac{2}{3} C_i \dot{\epsilon}^{pl} - \gamma_i \dot{\epsilon}^{pl} \alpha_i \quad (2)$$

where C_i and γ_i are model material parameters, $\dot{\epsilon}^{pl}$ is the plastic strain rate, and $\dot{\epsilon}^{pl}$ is the magnitude of the plastic strain rate. The Chaboche model parameters for Sn-3.0Ag-0.5Cu are detailed in Table 2.

The creep behaviour part of approach 1 was modelled with the Garofalo-Arrhenius constitutive law,

$$\dot{\epsilon}^{cr} = A[\sinh(B\sigma)]^n \exp\left(-\frac{Q}{RT}\right) \quad (3)$$

where $\dot{\epsilon}^{cr}$ is the steady-state creep strain rate, σ is the applied stress, T is the temperature in Kelvin, A and B are material constants, R is the gas constant, Q is an activation energy, and n is a stress exponent. The Garofalo-Arrhenius creep model reported in⁶⁰ was deployed with temperature-

dependant values of the material constant A to capture the creep behaviour of Sn-3.0Ag-0.5Cu. The creep constitutive law parameters are detailed in Table 3.

For approach 2, the Anand model constitutive law was used for the inelastic strain rate of Sn-3.0Ag-0.5Cu and the respective model constants were the same as those reported in ³ (Table 2 in that paper).

The finite element analysis (FEA) revealed that the displacement predictions for the solder joints in approach 1 (separate kinematic hardening plasticity and pure creep constitutive laws) were very closely matched to the equivalent simulation run with approach 2 (the Anand model with unified inelastic strain rate). Figures SI-Figure 2 and SI-Figure 3 (in the Supplementary Information) show a comparative analysis of the displacement results for solder joint M09 from the two macro-scale model approaches, showing that nodal displacements calculated from the two approaches are very similar, and the extreme point displacements in the solder joint differ by at most 3.7% and generally by <1%. Statistical analysis of the relative difference in predicted displacements for all solder joint mesh nodes, predicted with the two different material constitutive law approaches for Sn-3Ag-0.5Cu, show that the mean (average) value of the difference is only 0.8% and the respective standard deviation is 0.9%. This high similarity suggests that, in this study, kinematic hardening does not have a significant effect on the deformation of solder joints, and that displacement is dominated by creep. Several reasons for this are identified: (1) Sn-3.0Ag-0.5Cu has a high homologous temperature over the investigated mild temperature cycle 0°C to 100°C ($T/T_m = 0.54 - 0.74$) which makes creep behaviour strongly dominant over kinematic hardening plasticity, (2) the long temperature cycle duration of 40 minutes, with 10-minute dwells at both the low (0°C) and the high (100°C) temperature extreme, is associated with low strain rates that favour creep deformation and stress relaxation throughout the cycle, (3) solder joint movements at the component assembly level are kinematically constrained by the movement of the printed circuit board and the BGA package. While these conclusions have been drawn for the problem in this paper, solder joints subjected to temperature cycles with lower-temperature regimes and fast cycling rates could promote kinematic hardening behaviour over creep. Due to the similar results from the two macro-scale model approaches here, only the Anand model transient displacement boundary conditions were passed over to the microstructure level CPFEE model.

Table 2. Chaboche kinematic hardening model parameters for Sn-3.0Ag-0.5Cu

Temp [°C]	σ_0 [MPa]	C_1	γ_1	C_2	γ_2	C_2	γ_2
-40	36.46	49594	5200	24330	1485	984	211
25	24.96	27809	5162	13481	1476	548	211
75	19.05	18714	5140	9185	1473	376	210
125	14.64	12915	5145	6502	1474	268	210

Table 3. Garofalo-Arrhenius creep strain rate model parameters for Sn-3.0Ag-0.5Cu

Temp [°C]	A [s^{-1}]	B [MPa^{-1}]	n [–]	Q/R [K^{-1}]
25	31974			
75	79512			
100	134720	0.02447	6.41	6498.32
125	213760			

SI-Figure 2 Solder joint M09 contour plots of displacement (UX, UY and UZ) at the high-temperature extreme of the thermal cycle (100°C) predicted under two different setups for the Sn-3.0Ag-0.5Cu material behaviour: (1) Chaboche (kinematic hardening rule) and Garofalo (creep), and (2) Anand viscoplastic model. Peak values of displacements and their percentages difference with the two model setups are tabulated under the contour plots.

SI-Figure 3: Transient displacement results (UX, UY and UZ) of the central point at top interface (P1) and bottom interface (P2) of solder joint M9, predicted under two different setups for the Sn-3.0Ag-0.5Cu material behaviour: (1) Chaboche kinematic hardening rule) and Garofalo (creep), and (2) Anand viscoplastic model. The top centre graph details the profile of the simulated isothermal load, representing the post-reflow cooldown of the assembly (causing residual stress) followed by five temperature cycles of 0-100°C.”

REVIEWERS' COMMENTS

Reviewer #1 (Remarks to the Author):

The authors have thoroughly revised their manuscript, thank you for all your hard work.

This reviewer can accept the manuscript in its current form.

Reviewer #3 (Remarks to the Author):

Thank you for your careful answering and revising.

You have compared the FEA result between the result using the Anand model and that using the Chaboche model combined with the Garofalo creep law and showed that the difference was small. Although the displacement boundary conditions derived from the FEA using the above-combined model should be passed to the micro-scale FEA, the comparison you did almost resolved my concern.